

# Feasibility of continuous physical activity monitoring: first-month recovery markers following joint replacement surgery

Arash Ghaffari, Andreas Kappel, Thomas Jakobsen, Søren Kold and Ole Rahbek

Orthopaedic Surgery Department, Aalborg University Hospital, Aalborg, Denmark

## ABSTRACT

**Background**. The role of physical activity (PA) monitoring during the recovery after orthopaedic surgeries is unclear. This study aimed to explore early changes in the pattern and level of PA following orthopedic surgeries.

**Methods**. This observational feasibility study included 11 hip replacement patients (four females) with a mean age of 66 years and five knee replacement patients (four females) with a mean age of 65 years. A PA tracker was used to collect the patients' daily PA data, including duration of various activity categories, steps taken, and activity intensity count. The PA tracker recorded data from two weeks prior to surgery until four weeks after the surgery. Ratios of PA parameters for each of the first four weeks following surgery were calculated in relation to preoperative measurements.

**Results**. Compared to preoperative measurements, during the first four weeks after the surgery, the time spent in a recumbent position decreased from 112% to 106%, while continuous walking time and activity intensity count increased from 27% to 77% and from 35% to 73%, respectively. Step counts increased from 18% to 67%, and sit-to-stands rose from 65% to 93%. No significant changes were found in sitting, standing, sporadic walk time, and sporadic steps.

**Conclusion**. Continuously measuring PA using wearable sensors was feasible in orthopedic patients during the perioperative period. Continuous step count, walking time, activity intensity count showed noticeable changes and stable patterns demonstrating their potential for remote monitoring of patients during the early postoperative period.

## INTRODUCTION

Orthopedic procedures and especially joint replacement surgeries require careful postoperative observation to ensure optimal patient recovery. Adjusting treatment plans based on real-time patient data is crucial for facilitating this recovery process (*Whiting et al., 2015*). However, the conventional method of regular follow-up visits scheduled at fixed intervals and conducted in hospital outpatient clinics often proves inconvenient for patients and may fail to timely identify issues in the patient's recovery process (*Gualandi et al., 2019*). To address these challenges, telemedicine has emerged as a promising solution,

Corresponding author
Arash Ghaffari, a.ghaffari@rn.dk

leveraging technological advancements to enable more frequent and remote monitoring of patients (*Bahadori et al., 2020*).

This innovative approach is particularly relevant in the field of orthopedics, where wearable technology has gained considerable attention. Wearable sensors allow continuous monitoring and enable objective data collection on various aspects of physical activity (PA), reducing reliance on potentially biased self-reports and providing a comprehensive view of a patient's activity levels and recovery progress (*Wainwright & Kehlet, 2022*). Continuous monitoring facilitates the early detection of postoperative complications, allowing timely interventions (*Master et al., 2022*). Additionally, data from wearable devices can inform personalized rehabilitation programs, tailoring exercises to individual needs (*Rodgers et al., 2019*). Wearable technology also enhances patient engagement by providing real-time feedback and encouraging compliance with rehabilitation protocols (*Bahadori, Immins & Wainwright, 2018*).

To effectively utilize PA as a functional indicator for postoperative recovery, it is crucial to have a comprehensive understanding of the dynamic nature of both the quantity and quality of PA following surgery. Unlike simple indicators such as heart rate or body temperature, PA provides a more holistic assessment of the rehabilitation and functional recovery process, integrating both the intensity and pattern of movements (*Arnold, Walters & Ferrar, 2016*). However, the interpretation and reliability of PA data are complex and influenced by factors including postoperative pain, medications, mood, and individual motivation (*Sim et al., 2022*). While subjective patient expectations can affect perceived recovery outcomes, the objective measurements provided by PA trackers offer a reliable and unbiased assessment of PA levels during the postoperative period.

Previous studies have explored the role of sensor-based PA monitoring in orthopaedic surgeries, particularly arthroplasties (*Iovanel, Ayers & Zheng, 2023*; *Hammett et al., 2018*). However, these investigations often lacked continuous assessment, focusing instead on discrete postoperative intervals (*Gianzina et al., 2023*; *Sosdian et al., 2014*; *Boekesteijn et al., 2022*). Feasibility studies highlight the potential of wearable sensors in various clinical settings, stressing the importance of evaluating practical aspects such as patient adherence, data accuracy, and overall acceptability (*Lebleu et al., 2021*; *Luna et al., 2019*). However, a significant research gap exists in understanding how wearable technology can be employed for continuous monitoring of postoperative recovery, especially during the immediate postoperative period when patients are most vulnerable to complications (*Natarajan et al., 2023*). Therefore, further investigation is warranted to fully explore the capabilities of wearable technology in monitoring postoperative recovery and its potential as a valuable tool in assessing the nuanced changes in PA during this critical phase.

This study is an observational feasibility study aimed at assessing the practicality and efficacy of continuously measuring PA using wearable sensors in patients who underwent total knee arthroplasty (TKA) and total hip arthroplasty (THA). Our primary objective was to evaluate the feasibility of utilizing wearable sensors to capture continuous PA data during the perioperative period. Additionally, we examined the changes in PA patterns and levels following orthopedic surgeries in a detailed and comprehensive manner. We analyzed the variations in the quantity of different daily PAs during this early postoperative period,

evaluated their patterns, and identified the most predictable, consistent, and valuable variables. Through this feasibility study, we aimed to provide insights into the utilization of wearable sensors for monitoring PA in orthopedic surgery patients, with a specific focus on the early postoperative period.

## MATERIALS & METHODS

### Study design and setting

In this observational feasibility study, the PA data analyzed covered a period of two weeks before the surgery and continued for four weeks after the surgical procedure. The study was registered in the North Jutland research database in Denmark (registration ID: 2021-119) and ethical approval was not deemed necessary for this study in accordance with the request made to the region North Jutland scientific ethics committee (ref. no. 2021-000438). The article complied with the guidelines for Strengthening the Reporting of Observational Studies in Epidemiology (STROBE) (*Von Elm et al., 2007*).

### Participants

The patients included in this feasibility study were part of a larger cohort of patients enrolled in another study aimed at exploring the use of wearable sensors for monitoring PA in orthopedic patients (*Ghaffari et al., 2023a*). For this specific feasibility assessment, a subset of patients was chosen to ensure a more homogeneous group in terms of the operation (lower limb joint replacement) to enhance the reliability of continuous PA monitoring during the perioperative period.

Patients were selected based on their scheduled hip or knee replacement surgeries at Aalborg University Hospital between November 2021 and July 2022. Inclusion criteria required a minimum waiting period of two weeks before surgery. Patients who had undergone any other lower limb surgery within the previous six months were excluded. Additionally, patients were excluded if they were unable to walk independently or lacked the capability to use the PA tracker effectively. The capability to use the PA tracker was assumed to be sufficient in patients who were already using smartphones, as operating the PA tracker primarily involved opening an app to transfer data. This assumption was based on the minimal technical requirements needed to use the tracker effectively. All eligible participants were thoroughly informed about the study through both written materials and verbal communication, and informed consent was obtained using electronic forms on the REDCap platform.

### Variables

Basic and demographic information, including age, sex, height, weight, and medical and surgical history was registered in a REDCap database hosted by Region North Jutland, Denmark. To obtain subjective information regarding the patients' level of PA before the surgery, we utilized the International Physical Activity Questionnaire (IPAQ) (*Craig et al., 2003*). The IPAQ short form utilized in this study evaluates PA in various domains through inquiries about the frequency (measured in days per week) and duration (time per day) of three distinct types of activities (walking, moderate-intensity activities, and vigorous-intensity activities) performed in these domains (leisure time, domestic and gardening

activities, work-related and transport-related activity). Accordingly, the participants were classified into three categories based on their PA levels: inactive, minimally active, and health-enhancing physically active (HEPA) (*Craig et al., 2003*).

- **Inactive:** Participants were classified as inactive if they reported no activity or some activity but not enough to meet the criteria for the minimally active category.
- **Minimally Active:** Participants were classified as minimally active if they engaged in physical activities sufficient to meet any of the following criteria:
  - At least 3 days of vigorous-intensity activity totaling at least 20 minutes per day.
  - At least 5 days of moderate-intensity activity or walking totaling at least 30 minutes per day.
  - At least 5 days of any combination of walking, moderate-intensity, or vigorous-intensity activities.
- **Health-Enhancing Physically Active (HEPA):** Participants were classified as HEPA if they met the following criteria:
  - Vigorous-intensity activity on at least 3 days.
  - Any combination of walking, moderate-intensity, or vigorous-intensity activities on at least 7 days.

## PA trackers

PA measurement was conducted using the SENS Motion® PA tracker (Copenhagen, Denmark), which comprises a triaxial accelerometer sampling at a frequency of 12.5 Hz with a range of $\pm 4$G. These PA trackers have demonstrated high reliability and validity in various studies. *Pedersen et al. (2022)* validated the device in slow- and fast-walking hospitalized patients, reporting a 95% accuracy rate for step counts. Similarly, *Bartholdy et al. (2018)* confirmed the reliability of the PA tracker in detecting sedentary behavior with an accuracy rate of approximately 90%. The criterion validity for measuring linear accelerations during overground walking was established with a strong correlation ($r = 0.89$) with gold-standard gait analysis systems (*Ghaffari et al., 2022*).

While the sensors did not require calibration, there were potential sources of error, such as positioning errors, data transmission issues, environmental factors, and user compliance. To minimize errors related to incorrect positioning of the tracker, patients received detailed instructions on how to wear the device, and during the preliminary session, they were supervised to ensure correct placement. Any issues with data transmission were promptly addressed by contacting the patients, as the devices utilized Bluetooth technology to securely transmit data to a web server *via* patients' smartphones.

Environmental factors such as interference from electronic devices were mitigated by advising patients to avoid placing the tracker near such devices. To ensure continuous data collection, patients were instructed to wear the PA tracker at all times, except during activities that could damage the device (*e.g.*, swimming). Compliance was monitored, and reminders were sent if data transmission was delayed by more than 72 hours.

The SENS Motion algorithm transforms the raw accelerations into different PA categories (*Bartholdy, 2019*):

- The time spent on different types of PA:

  – Recumbent (lying down)
  – Sitting
  – Standing
  – Continuous walking
  – Sporadic walking

- The number of:

  – Sit-to-stands
  – Step counts during regular continuous, sporadic, and slow continuous walking

- The activity intensity count, which is a measure based on the magnitude of accelerations and indicates the intensity of PA.

The PA trackers were attached to the patient's lateral distal thigh using a special Band-Aid (Medipore™, 3M, Soft Cloth Surgical Tape on Liner). These trackers were placed on the side opposite the planned surgical site. Throughout the study, the patients constantly wore the trackers and utilized their smartphones' Bluetooth to securely transmit the data to a web server.

## Data analysis

We conducted the data analysis in several steps to ensure a comprehensive understanding of the PA patterns and changes. The steps are as follows:

1. Time series visualization:

   We created time series plots to visualize the overall trend in each PA category (*e.g.*, recumbent time, sitting time, continuous walking time) for both the TKA and THA groups. These plots provided a preliminary visual understanding of how PA levels changed over time from the preoperative period to the postoperative period.

2. Calculation of mean values:

   We calculated the mean values for each PA variable for the preoperative period and each of the four postoperative weeks. To standardize the comparison, the average value for the preoperative period was considered as 100%, and the ratios of values for each postoperative week to the preoperative value were calculated. This allowed us to quantify the relative change in PA levels over time.

3. Statistical comparison:

   To assess the significance of changes in PA levels, we compared the average preoperative values with the postoperative values for each week. This involved consecutive comparisons: preoperative *vs.* postoperative week one, week one *vs.* week two, week two *vs.* week three, and week three *vs.* week four. The Wilcoxon signed-rank test was used for these comparisons due to the paired nature of the data and the small sample size.

4. Autocorrelation analysis:

   To determine whether there was a consistent pattern of change in each PA category, we used the autocorrelation function (ACF) (*Shumway & Stoffer, 2017*). ACF is a statistical

measure that quantifies the similarity between a time series and a lagged version of itself over successive time intervals.

This method was chosen because it offers valuable insights into the underlying temporal patterns within the data and helps identify consistent patterns of change in each PA category over time (*Mitchell et al., 2020*). Autocorrelation coefficients help us quantify the strength and direction of the relationship between PA values at different time lags. A significant positive autocorrelation coefficient indicates dependence between the values of the PA category, suggesting a consistent pattern of change over time. In contrast, a weak coefficient implies that the values of the time series at different time points are less dependent on each other, indicating that the time series is more random or noisy.

Before utilizing autocorrelation, we preprocessed the daily PA data by computing the rolling median with a three-day window size to minimize the influence of anomalies in the data. We then computed the average autocorrelation coefficients for lags of one to seven days across all subjects for each PA category. The results were illustrated using a heat map, depicting the relationship between day lags and PA categories, thereby providing a clear visual representation of the temporal dependencies in the PA data.

## Statistical methods

Descriptive statistics were used to summarize the characteristics of patients in the TKA and THA groups. The Mann–Whitney test was used to compare the numerical variables, including age, BMI, height, and weight. The Fisher Exact test was used to compare the categorical variables, including age and the number of patients with medical comorbidities. The level of significance was set at $p < 0.05$.

The Wilcoxon signed-rank test was used to compare the PA values of the preoperative period and postoperative weeks one, two, three, and four. This test was conducted on paired data for each participant.

Since this was a feasibility study, a convenient sample size was selected to explore the potential of using wearable sensors to measure PA during the perioperative period in patients undergoing TKA and THA. The primary aim was to assess the practicality and efficacy of continuous PA monitoring using wearable sensors, rather than to provide conclusive statistical evidence. Therefore, a smaller sample size was deemed sufficient for evaluating feasibility and generating preliminary data to inform future, larger-scale studies.

The datasets generated and analyzed during the current study are available in the Figshare repository, (doi: https://doi.org/10.6084/m9.figshare.26534227).

## RESULTS

The characteristics of the patients are demonstrated in Table 1. Thirteen out of the 16 patients in the study had previous lower limb surgeries performed more than six months before the current surgery, with previous surgeries including contralateral hip ($n = 6$) and knee arthroplasty ($n = 3$), osteotomies around the knee ($n = 2$), and other lower limb surgeries ($n = 2$). All included patients underwent primary arthroplasties in the knee or hip joints, and the surgical procedures and postoperative periods were without complications.

**Table 1  Participants' demographic data.**

| Variable | | Knee replacement group ($n = 5$) | Hip replacement group ($n = 11$) | All patients ($n = 16$) |
|---|---|---|---|---|
| Age (yrs) | Median [Range] | 65 [25–74] | 66 [43–78] | 66 [25–78] |
| | <50 ($n$ (%)) | 1 (20) | 4 (36) | 5 (31) |
| | 50–70 ($n$ (%)) | 2 (40) | 4 (36) | 6 (38) |
| | 70 < ($n$ (%)) | 2 (40) | 3 (27) | 5 (31) |
| Sex | Female ($n$ (%)) | 4 (80) | 4 (36) | 8 (50) |
| | Male ($n$ (%)) | 1 (20) | 7 (64) | 8 (50) |
| Medical comorbidity | Cardiac Disease ($n$ (%)) | 1 (20) | 2 (18) | 3 (19) |
| | Asthma/Allergy ($n$ (%)) | 1 (20) | 0 | 1 (6) |
| | Hypertension ($n$ (%)) | 0 | 3 (27) | 3 (19) |
| | Diabetes Type 2 ($n$ (%)) | 0 | 1 (9) | 1 (6) |
| | Total ($n$ (%)) | 2 (40) | 6 (55) | 8 (50) |
| Lower limb surgery history ($n$ (%)) | | 5 (100) | 8 (73) | 10 (81) |
| Height (cm) (median [Range]) | | 165 [142–180] | 177 [156–190] | 174 [142–190] |
| Weight (kg) (median [Range]) | | 71 [39–88] | 94 [62–146] | 85 [39–146] |
| BMI[a] (kg/m$^2$) (median [Range]) | | 26.1 [19.3–29.0] | 27.8 [25.0–42.7] | 27.5 [19.3–42.7] |
| IPAQ[b] | Inactive ($n$ (%)) | 2 (40) | 4 (36) | 6 (38) |
| | Minimally active ($n$ (%)) | 2 (40) | 6 (55) | 8 (50) |
| | HEPA[c] active ($n$ (%)) | 1 (20) | 1 (9) | 2 (13) |

Notes.
[a]Body mass index.
[b]International Physical Activity Questionnaire.
[c]Health-enhancing physical activity.

A dataset comprising 660 days of data from 16 patients was used for analysis. Complete data consisting of 14 days of preoperative data and 28 days of postoperative data were available for 11 participants, and five patients had slightly deviated data availability. Specifically, two patients had 12 days of preoperative data, while another had 13 preoperative days of data. Additionally, one patient provided 26 days of postoperative data, and another provided 23 days. These minor variations in data availability were considered during subsequent analyses and interpretation of the study's findings.

## Feasibility outcomes

**Data collection compliance:** The overall compliance rate was high, with minimal gaps in data. Despite the missing 12 days of data, most patients provided complete data sets for both preoperative and postoperative periods.

**Wearability and comfort:** Patients reported no significant discomfort while wearing the PA trackers. No adverse events related to the use of the devices were reported, indicating good tolerability.

**Data transmission and technical issues:** Data transmission *via* Bluetooth to the web server was successful for most of the patients. Issues with data transmission were promptly addressed by contacting the patients, and there were no significant disruptions in data flow.

Figure 1 illustrates the variations in different PA categories throughout the perioperative period, providing a visual representation of how PA levels fluctuate before and after surgery.

Table 2 displays the median and interquartile range (IQR) of various PA categories during the preoperative period and four weeks after surgery, along with the corresponding *p*-values for comparing consecutive periods.

Additionally, Fig. 2 features a heatmap visualizing the average autocorrelation coefficients for various PA categories across all study subjects at different day lags. The heatmap highlights significant patterns, particularly showing that certain PA categories, such as continuous walking time, activity intensity count, and regular and slow continuous step counts, exhibit more consistent and stable changes over time compared to others.

## DISCUSSION

This study supports the practicality and effectiveness of wearable sensors in continuously monitoring PA in patients undergoing TKA and THA. The feasibility of using these sensors to collect continuous PA data throughout the perioperative period was demonstrated, with significant changes observed in key PA variables—such as step counts, continuous walking duration, and activity intensity count—during the critical early postoperative phase. These findings underscore the potential of wearable sensors to provide real-time, objective insights into patient recovery.

To ensure a robust baseline for postoperative comparisons, we collected wearable sensor data consistently over a two-week period before each subject's surgery. Although the literature suggests that 3 to 7 days of PA monitoring is generally sufficient to provide reliable estimates of habitual activity levels, even in patients with reduced mobility (*Ward et al., 2005*; *Hilden et al., 2023*; *Hart et al., 2011*), we extended this period to 14 days. This extended monitoring period was specifically chosen to better capture daily variations and mitigate potential biases introduced by preoperative factors such as pain and reduced range of motion (ROM). We continued PA data collection for four weeks post-surgery to identify early recovery trends. This process confirmed the feasibility of collecting such data, revealing significant changes in PA, particularly in continuous walking metrics like step count and walking duration. A notable decrease in these variables was observed immediately after surgery, followed by a gradual increase over the first four weeks, suggesting their value as recovery indicators. Consistently, activity intensity counts corroborated these findings. Aligning with previous studies (*Crizer et al., 2017*), we noted changes in moderate PA and walking post-surgery, though these studies typically report outcomes months post-surgery (*Harding et al., 2014*; *Lin et al., 2013*; *Lützner, Kirschner & Lützner, 2014*). In a recent study, *Christensen et al. (2023)* examined a large cohort undergoing TKA and identified significant post-surgical improvements in various PA tracker variables, highlighting gait parameters as crucial recovery markers.

We employed autocorrelation to uncover temporal patterns in the time series data, which facilitates comprehending postoperative recovery dynamics and recognizing stable recovery markers. The strong short-term autocorrelation in categories like continuous walking time, activity intensity count, regular continuous steps, and slow continuous steps

Peer

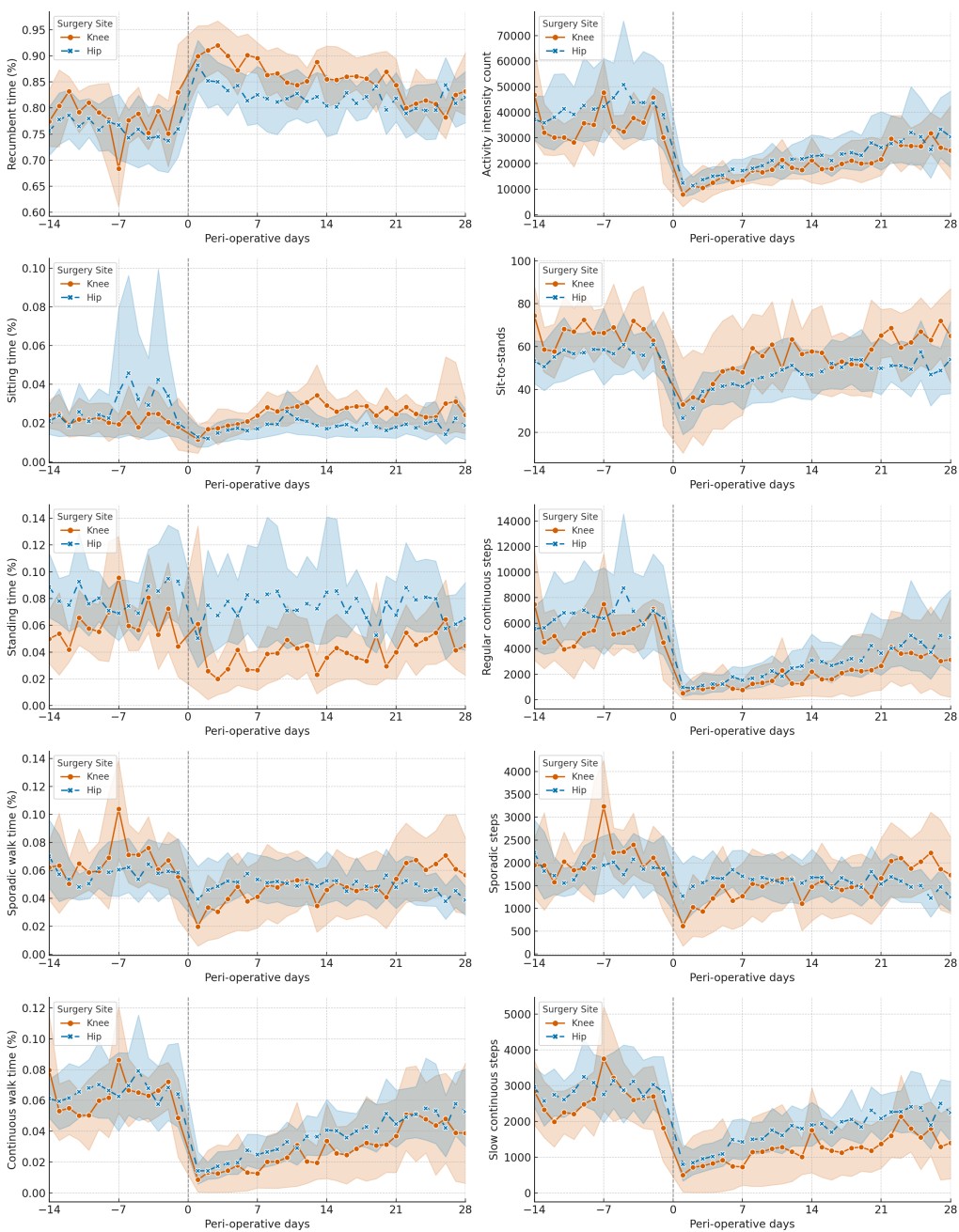

**Figure 1** **Time-series for different physical activity (PA) categories for total knee arthroplasty (TKA) and total hip arthroplasty (THA) groups.** The trend of daily changes in average physical activities for the TKA and THA groups is represented in orange and blue, respectively. The shaded areas represent the variability within the groups, illustrating the interquartile range (IQR) of the data.

indicate stable and predictable PA patterns in the immediate postoperative period. This consistency reflects adherence to rehabilitation protocols, emphasizing regular, controlled walking to enhance recovery, build endurance, and prevent complications such as muscle

atrophy and joint stiffness (*Ward et al., 2021*). However, it is important to recognize that while our study shows positive recovery trends, overall progress appears longer than anticipated. Full recovery often extends well beyond the first four weeks post-surgery, with many patients not returning to their baseline PA levels (*Harding et al., 2014*; *Rivera et al., 2024*; *Bin sheeha et al., 2020*). This highlights the challenges of early recovery, including persistent pain, reduced mobility, and the psychological effects of surgery—all of which can delay the return to normal activity levels. The discrepancies between patient-reported outcomes (PROMs) and objective PA measurements further underscore the need to integrate objective assessments, such as those provided by wearable sensors, with PROMs to achieve a more comprehensive understanding of recovery trajectories (*Wainwright & Kehlet, 2022*).

We observed no significant postoperative changes in daily sitting and standing times compared to the preoperative period. This lack of variation suggests that these variables might not effectively capture the nuances of early recovery. Interestingly, we also found no significant change in sporadic walking and the number of sporadic steps. This limitation could be attributed to the current sensor technology, which tends to be more accurate in detecting continuous walking patterns rather than sporadic movements. Wearable sensors may struggle with identifying irregular, brief periods of activity, potentially leading to underreporting or inaccuracies in these data points. Studies, particularly in populations with conditions like Parkinson's disease, have similarly shown reduced accuracy in capturing sporadic walking (*Salarian et al., 2004*; *Nguyen et al., 2017*; *Caldas et al., 2017*). Addressing these limitations will be crucial for the development of future sensor technologies. Potential improvements might include refining the algorithms that detect short bursts of activity or enhancing the sensitivity of the devices to better differentiate between various types of movement (*Yang et al., 2022*).

While we noted no significant difference in sitting time during the initial postoperative week compared to the preoperative period, there was a notable increase in the second week, possibly attributed to increased bed rest, followed by a subsequent decrease. The time spent in a recumbent position was consistently higher post-surgery for the entire four-week period, albeit with some fluctuations, indicating areas for further exploration. Interestingly, sit-to-stand movements decreased post-surgery but then significantly increased in the first two weeks, eventually stabilizing. This trend was more consistent in comparison to sitting time. Sit-to-stand, a critical mobility metric, was observed in frequencies within the range reported in existing literature (*Bohannon, 2015*). Understanding the frequency of sit-to-stand movements can be crucial in identifying recovery challenges.

This study pioneers in assessing both the quantity and pattern of continuous PA data using a small wearable sensor as an early marker of recovery post-orthopedic surgery. However, it has several limitations. It is important to acknowledge the small sample size and the potential for generalization issues between TKA and THA patients. The rate of recovery can vary significantly between these groups, and drawing broad conclusions on detailed recovery is challenging with the limited number of participants in each subgroup. Notably, the inclusion of a 24-year-old woman with achondroplasia in the TKA group, diverging from the typical TKA patient demographic, was a deliberate choice due to the study's

Ghaffari et al. (2024), *PeerJ*, DOI 10.7717/peerj.18285

**Table 2  Physical activities during the perioperative period.** Median and interquartile range (IQR) of physical activity categories during the preoperative period and four weeks postoperatively, with *p*-values for comparing consecutive periods.

| Variable[a] | Preop (*n* = 16) | Post-op Week 1 (*n* = 16) | Preop-Week 1 *p*-value | Post-op week 2 (*n* = 16) | Week 1–Week 2 *p*-value | Post-op week 3 (*n* = 16) | Week 2–Week 3 *p*-value | Post-op Week 4 (*n* = 16) | Week 3–Week 4 *p*-value |
|---|---|---|---|---|---|---|---|---|---|
| Recumbent time (%) | 75 [72, 82] | 86 [84, 89] | .0003 | 84 [78, 87] | .005 | 83 [78, 86] | .6 | 79 [77, 87] | .005 |
| Sitting time (%) | 1.6 [1.3, 2.2] | 1.6 [1.1, 2.0] | .1 | 2.1 [1.5, 3.0] | .0002 | 2.0 [1.4, 2.8] | .03 | 1.9 [1.2, 2.7] | .8 |
| Standing time (%) | 6.8 [5.2, 8.9] | 5.4 [2.8, 7.0] | .1 | 5.2 [3.5, 7.5] | .08 | 5.5 [4.6, 7.4] | .9 | 6.2 [4.6, 8.2] | .08 |
| Sporadic walk time (%) | 5.5 [4.4, 7.8] | 4.3 [2.8, 5.5] | .02 | 4.7 [3.8, 6.1] | .1 | 4.6 [3.6, 6.2] | .3 | 5.0 [3.4, 5.8] | .2 |
| Continuous walk time (%) | 6.4 [4.1, 7.5] | 0.017 [0.006, 0.027] | <.0001 | 0.027 [0.011, 0.045] | .0003 | 0.037 [0.024, 0.042] | .003 | 0.04 [0.032, 0.064] | .0001 |
| Activity intensity count (×1000) | 36 [30.5,41.2] | 13.3 [9.3,16.6] | <.0001 | 18.5 [13.6,25.0] | .0004 | 21.0 [17.5,23.3] | .04 | 24.9 [20.6,33.2] | <.0001 |
| Sit-to-stand (*n*) | 60 [50, 71] | 35 [30, 48] | .0008 | 51 [37, 66] | .005 | 50 [40, 65] | .6 | 56 [43, 64] | .2 |
| Regular continuous steps (*n*) | 5784 [3495, 6991] | 821 [269, 1671] | <.0001 | 1819 [557, 3052] | .001 | 2712 [1468, 2909] | .006 | 2940 [1914, 5304] | <.0001 |
| Sporadic steps (*n*) | 1782 [1460, 2449] | 1352 [940, 1858] | .02 | 1594 [1239, 1909] | .1 | 1553 [1123, 1958] | .3 | 1640 [1126, 1848] | .2 |
| Slow continuous steps (*n*) | 2609[2161, 3391] | 880 [406, 1387] | <.0001 | 1509 [787, 2221] | .0004 | 1900 [1191, 2240] | .08 | 2139 [1534, 2692] | .001 |

**Notes.**

[a] All variables correspond to daily values.
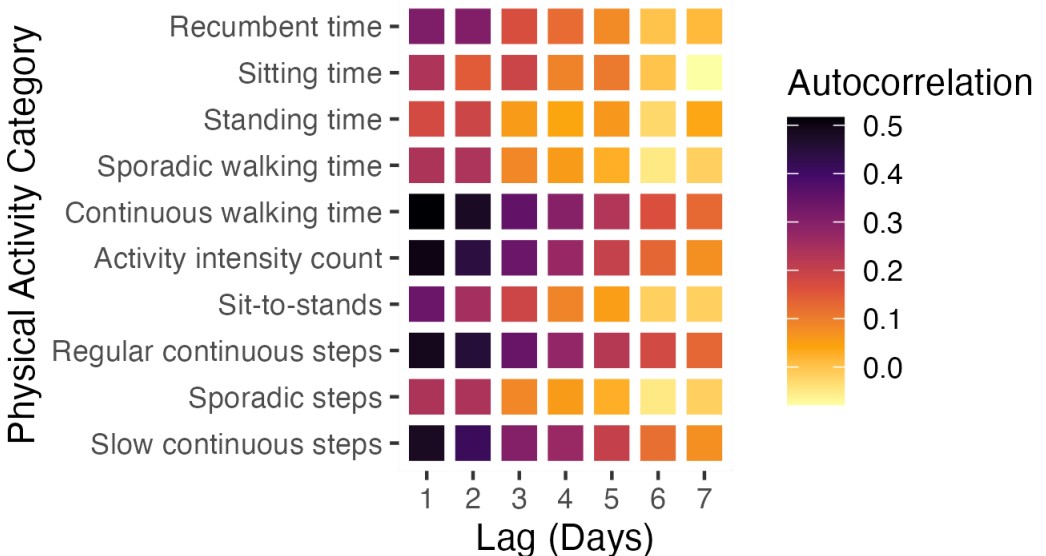

**Figure 2 Autocorrelation coefficients of different PA categories.** Heatmap summarizing the mean auto-correlation coefficients across all participants for different PA categories ($y$-axis) and day lags ($x$-axis).

focus on feasibility over comprehensive demographic representation. Future studies with larger sample sizes and separate analyses for TKA and THA patients are recommended to provide more definitive conclusions. Furthermore, the uneven distribution of participants between the TKA and THA groups precluded detailed statistical comparisons. While we did not validate the PA changes against established recovery assessment tools, our focus was to explore the feasibility of PA monitoring in the early postoperative phase. Despite the ongoing process of full validation, preliminary evidence supports the potential of wearable sensors in this context (*Bartholdy et al., 2018*; *Christensen et al., 2014*; *Ghaffari et al., 2023b*).

Our study demonstrates the feasibility of using small wearable sensors in collecting PA data during the perioperative period following orthopedic surgeries, marking a significant advancement in the realm of telemedicine for post-surgery care. The introduction of these sensors has transformed the collection and analysis of human motion data, offering a more convenient and precise approach compared to traditional in-person recovery monitoring methods. By tracking PA, these sensors provide essential insights into a patient's health and well-being during recovery (*Bohannon, 2015*; *Hendricks, Chong & Cusick, 2018*; *Warburton, Nicol & Bredin, 2006*; *Bowyer & Royse, 2016*). However, while PA data serves as a promising recovery marker due to its significant, stable, and predictable changes post-surgery, it is crucial to recognize that recovery may extend well beyond the early postoperative period (*Wainwright & Kehlet, 2022*). The full potential of this technology in broader, population-based studies remains to be explored. Future research should focus on larger-scale analyses to deepen our understanding of PA patterns in diverse patient groups and further establish PA as a reliable indicator of successful recovery.

## CONCLUSIONS

This study demonstrates the feasibility of using small wearable sensors to continuously and objectively track PA during the critical perioperative phase of orthopedic recovery. Specifically, variables such as step counts, continuous walking duration, and activity intensity count emerged as promising markers for remotely monitoring early recovery after TKA and THA surgeries. However, given the small sample size and the exploratory nature of this study, these findings should be interpreted with caution.

While our results offer preliminary insights into the potential utility of wearable sensors in postoperative care, further research with larger, more diverse populations is essential to validate these findings and refine the use of these markers in clinical practice.

## ACKNOWLEDGEMENTS

We extend our gratitude to Tina Lyngholm Jensen and Rikke Emilie Kildahl Lauritsen for their invaluable assistance in patient recruitment and inclusion in this study.

### Funding
The authors received no funding for this work.

### Competing Interests
The authors declare there are no competing interests.

### Author Contributions
- Arash Ghaffari conceived and designed the experiments, performed the experiments, analyzed the data, prepared figures and/or tables, authored or reviewed drafts of the article, and approved the final draft.
- Andreas Kappel conceived and designed the experiments, performed the experiments, authored or reviewed drafts of the article, and approved the final draft.
- Thomas Jakobsen conceived and designed the experiments, performed the experiments, authored or reviewed drafts of the article, and approved the final draft.
- Søren Kold conceived and designed the experiments, authored or reviewed drafts of the article, and approved the final draft.
- Ole Rahbek conceived and designed the experiments, authored or reviewed drafts of the article, and approved the final draft.

### Human Ethics
The following information was supplied relating to ethical approvals (*i.e.*, approving body and any reference numbers):

Ethical approval was not deemed necessary for this study in accordance with the request made to region North Jutland scientific ethics committee (ref. no. 2021-000438).

## Data Availability

The datasets are available at Figshare: Ghaffari, Arash; Kappel, Andreas; Jakobsen, Thomas; Kold, Søren; Rahbek, Ole (2024). Perioperative Physical Activity Monitoring Data—TKA and THA Patients. figshare. Dataset. https://doi.org/10.6084/m9.figshare.26534227.v3.

## Supplemental Information

Supplemental information for this article can be found online at http://dx.doi.org/10.7717/peerj.18285#supplemental-information.

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
