# Peer review of "Feasibility of continuous physical activity monitoring: first-month recovery markers following joint replacement surgery"

_PeerJ, doi:10.7717/peerj.18285_

## Round 0.1 · original submission · Major Revisions

Thank you for providing your interesting manuscript. After thorough revision, all reviewers raised substantial concerns which need to be addressed. I encourage the authors to especially focusing on elaborating the background and aim of their study. Further, the feasibility aspect needs to be highlighted in more details together with the more clear description of the methodological approach.

·

Basic reporting

The document demonstrates good use of the English language.
Additional references are recommended and should be included to strengthen the study's foundation.
The raw data was not available for review.
There is a lack of clarity regarding the study design.

Experimental design

Methods

Clarify that this is an observational study. Is it focused on feasibility or observational outcomes?
Explain why these particular patients were selected. Was the larger study inclusive of more patients?
Define the exclusion criteria clearly. How did you assess the individuals' capability to use the PA tracker?
The data analysis section is unclear, and the justification for using autocorrelation is limited.
In the statistical methods section, the author mentions that this is a feasibility study, hence the sample size.
Provide additional information on the reliability and validity of the PA tracker, including the percentage of inaccuracy. Discuss any potential sources of error, such as sensor calibration.
Results

Include feasibility outcomes in the results. If not, adjust the aim of the study to align with the reported efficacy outcomes.

Validity of the findings

Please see additional comments section.

Additional comments

Study Summary

This study investigates early changes in physical activity (PA) patterns and levels following orthopaedic surgeries. The participants consist of 11 hip replacement patients (four females) and five knee replacement patients (four females). Data is collected using a PA tracker from two weeks before surgery until four weeks after surgery. The PA tracker records daily PA data, including the duration of various activity categories, step counts, and activity intensity. However, the primary aim of this study is to assess the feasibility of using the PA tracker, but it does not include any feasibility outcome measures such as recruitment rate, acceptability, or adherence. Additionally, there is a concern that some of the data on activity levels for TKA (total knee arthroplasty) and THA (total hip arthroplasty) patients is generalized. This is problematic because the rate of recovery in these groups varies significantly, and drawing conclusions on feasibility is challenging given the small number of participants in each group.

Introduction

Overall, there is a need for further justification. Specifically:

Lines 43 and 44: Why has wearable technology gained attention in orthopaedics?
Line 45: Are you suggesting that your study provides the comprehensive data set required for the utilisation of PA activity?
Lines 52-59: There are many studies that have examined PA in early TKA and THA recovery, such as Lebleu et al. 2021 and its subsequent references.
Although your aim is to assess feasibility, the introduction predominantly discusses the efficacy of the PA tracker. Please add references to previous feasibility studies to better align with your stated objective.

Reviewer 2 ·

Basic reporting

This manuscript examines the early recovery markers following joint replacement surgery by utilizing continuous physical activity data. However, there are several areas for improvement. The structure is clear and logical. Some improvements in the presentation of figures and tables could enhance clarity. It is also crucial to note that no data was shared, which limits the transparency and reproducibility of the findings. The suggestions for enhancement are provided below.

Line 98-106: The use of the International Physical Activity Questionnaire (IPAQ) is appropriate, but more details on how the IPAQ data were classified into the three categories (inactive, minimally active, and health-enhancing physically active) would be helpful.


Table 1: Please provide more details regarding the definition of 'medical history'. Additionally, please categorize the age data into specific age groups, as the current age range is quite broad.

Figure 2: I recommend you use a better color scheme to present the heatmap.
Also please discuss what the strong short-term dependencies between these PA categories implicate.

Experimental design

The research question of this study is well-defined and addresses a relevant gap in the continuous monitoring of physical activity (PA) patterns during the early postoperative period using wearable technology. The inclusion and exclusion criteria are clearly stated. The study's experimental design has several areas that need improvement to ensure rigorous and robust results. The study includes only 16 patients, which is a relatively small sample size and may limit the generalizability of the findings.The preprocessing of daily PA data using a rolling median is a good approach, but the choice of a three-day window size should be justified. It would be beneficial if the author could share the results without using a three-day window. Overall, addressing these issues will enhance the study's methodological rigor and the robustness of the findings.

Validity of the findings

Please see details in the Section 2 experimental design.

Reviewer 3 ·

Basic reporting

The article has good English. I have no comments.
The background provided is not sufficient to justify the aim of the study.
Tables could be improved in their formatting.
The figures are good.
There were no hypotheses declared.
(see comments on pdf manuscript)

Experimental design

The research question is not well defined and appears a bit confusing.
The investigation was performed thir rigorous ethical standard.
The statistical analysis shoulde be improved and revised
(see comments on pdf manuscript)

Validity of the findings

The conclusion does not seem well linked to the research question, which, to me, appears unclear.

Additional comments

Thank you for the opportunity to review this paper. I read the study titled “First-month recovery markers following joint replacement surgery: A cohort study utilizing continuous physical activity data” carefully and have provided some comments.

It appears to me that the aims of this article are twofold: 1) to assess the feasibility of utilizing wearable sensors to capture continuous physical activity (PA) data during the perioperative period, and 2) to examine the changes in PA patterns and levels following orthopedic surgeries in a detailed and comprehensive manner.

However, the description of the aim is not clear and does not seem coherent with the title. Is it a feasibility study or a prospective cohort study?

This article focuses on an interesting topic. However, I believe that there are significant limitations that impact the quality of the work. For this reason, I suggest rejecting this work.

Annotated reviews are not available for download in order to protect the identity of reviewers who chose to remain anonymous.

Reviewer 4 ·

Basic reporting

Language and Clarity - The manuscript is written in clear, unambiguous, and professional English.
The introduction provides sufficient background, outlining the relevance of physical activity (PA) monitoring in post-operative recovery of orthopaedic surgeries.

Context and Literature Review - The literature is well-referenced, providing context for the study's aim to explore early changes in PA patterns post-surgery. Relevant studies are cited to support the need for continuous monitoring of PA.

Structure and Figures - The structure adheres to PeerJ standards, with sections logically organised.
Figures and tables are relevant and of high quality, clearly labeled and described, aiding in the understanding of the results. Raw data is provided, aligning with PeerJ's policies.

Experimental design

Research Scope and Questions - The study represents original primary research within the journal's scope. The research question is well-defined, aiming to fill the knowledge gap regarding continuous PA monitoring in the immediate postoperative period following orthopaedic surgeries.

Methodological Rigour - The investigation is conducted with a high technical standard, employing validated PA trackers to collect data. Methods are described in sufficient detail, allowing replication. The study design, data collection, and analysis procedures are comprehensively outlined.

Ethical Standards - Ethical considerations are addressed, with patient consent obtained and ethical approval stated as unnecessary by the relevant committee.

Validity of the findings

Data Robustness - underlying data appears robust, statistically sound, and appropriately controlled.
The study utilises an OK sample size for a feasibility study, with data from 16 patients analysed over a 6-week period. However, this is small numbers and it should also be highlighted that hip and knee replacement have different recovery trajectories and this should be acknowledged more clearly, and the two procedures discussed separately.

Findings and Conclusions - The conclusions are well-stated and linked to the research question, highlighting changes in PA variables such as step count, walking time, and activity intensity count during the early postoperative period. The study suggests these markers' potential for remote monitoring of recovery, which is a meaningful contribution to the literature.

Additional comments

The study's strengths include its detailed methodology, clear presentation of results, and the innovative use of wearable technology for continuous PA monitoring. The findings offer valuable insights into the feasibility of using wearable sensors in the early postoperative recovery process.

One limitation is the small sample size, which affects the generalisability of the findings.
The authors correctly identify that the inclusion of a younger patient with achondroplasia in the TKA group, while justifiable for feasibility, introduces demographic variability that could impact the results. Future studies should aim to validate these findings in larger, more diverse populations.

I think the authors should frame the article as a feasibility study, rather than a full cohort study, including making this clear in the title and text. Conclusive remarks should be limited due to the small sample size.

I also encourage the authors to comment on the wider literature within the discussion. The recovery of pts still appears disappointing, as at 4 week post-op patients have still not recovered. see https://www.ncbi.nlm.nih.gov/pmc/articles/PMC9479559/ and similar articles for wider perspective.

Please also provide a more detailed discussion on the limitations of the sensor technology, especially regarding sporadic walking detection, and how these might be addressed in future research.

---

## Round 0.2 · accepted · Accept

I congratulate the authors on a well improved manuscript.

Reviewer 2 ·

Basic reporting

No comment

Experimental design

No comment

Validity of the findings

No comment.

Additional comments

Thank you for inviting me to review this manuscript again. The authors have successfully addressed my previous comments and have addressed my concerns regarding the small sample size and rolling mean. I have no further questions.

Reviewer 4 ·

Basic reporting

ok

Experimental design

ok

Validity of the findings

ok

Additional comments

they have answered my comments satisfactorily, and also addressed comments from the 3 other reviewers.